

# High tempo music prolongs high intensity exercise

Meaghan E. Maddigan, Kathleen M. Sullivan, Israel Halperin, Fabien A. Basset and David G. Behm

School of Human Kinetics and Recreation, Memorial University of Newfoundland, St. John's, NL, Canada

## ABSTRACT

Music has been shown to reduce rating of perceived exertion, increase exercise enjoyment and enhance exercise performance, mainly in low-moderate intensity exercises. However, the effects of music are less conclusive with high-intensity activities. The purpose of this with-participant design study was to compare the effects of high tempo music (130 bpm) to a no-music condition during repeated high intensity cycling bouts (80% of peak power output (PPO)) on the following measures: time to exercise end-point, rating of perceived exertion (RPE), heart rate (HR), breathing frequency, ventilatory kinetics and blood lactate (BL). Under the music condition, participants exercised 10.7% longer ($p = 0.035$; Effect size (ES) = 0.28) (increase of 1 min) and had higher HR (4%; $p = 0.043$; ES = 0.25), breathing frequency (11.6%; $p < 0.001$; ES = 0.57), and RER (7% at TTF; $p = 0.021$; ES = 1.1) during exercise, as measured at the exercise end-point. Trivial differences were observed between conditions in RPE and other ventilatory kinetics during exercise. Interestingly, 5 min post-exercise termination, HR recovery was 13.0% faster following the music condition ($p < 0.05$) despite that music was not played during this period. These results strengthen the notion that music can alter the association between central motor drive, central cardiovascular command and perceived exertion, and contribute to prolonged exercise durations at higher intensities along with a quicken HR recovery.

## INTRODUCTION

Music has long been thought to affect the senses (*Szmedra & Bacharach, 1998*) and can act as dissociation during exercise and thus enhance exercise enjoyment and performance (*Karageorghis & Priest, 2012a*, *2012b*; *Karageorghis, 2016*). Music is able to promote ergogenic and psychological benefits during exercise due to three proposed explanations (*Karageorghis & Priest, 2012a*, *2012b*; *Karageorghis, 2016*). First, music may allow individuals to separate thoughts from feelings. This divergence can change ones perception of unpleasant feelings, narrowing the performers attention, and reducing the sensations of fatigue during exercise (*Atkinson, Wilson & Eubank, 2004*; *Edworthy & Waring, 2006*; *Yamashita et al., 2006*; *Murrock & Higgins, 2009*). Second, the divergent stimulus (i.e., music) can alter psychomotor arousal (movement or muscular activity associated with

Corresponding authors
Israel Halperin,
Israel.halperin@mun.ca
Fabien A. Basset, fbasset@mun.ca

mental processes) and therefore can act as either a stimulant or a sedative prior to and during physical activity (*Bigliassi et al., in press*; *Carmichael et al., 2018*; *Szmedra & Bacharach, 1998*; *Yamamoto et al., 2003*; *Schücker et al., 2009*). The third explanation postulates that during continual submaximal activity, an individual is predisposed to respond to rhythmical elements (*Nikol et al., 2018*; *Terry et al., 2012*; *Waterhouse, Hudson & Edwards, 2010*); the result being synchronization between the tempo and the performer's movement making physical activity or exercise a more harmonious or less stressful experience (*Nikol et al., 2018*; *Rendi, Szabo & Szaba, 2008*; *Waterhouse, Hudson & Edwards, 2010*).

The available evidence on this topic is congruent and demonstrates that music can and does have a consistent and measurable effect on attention, the ability to trigger a range of emotions, affect mood, increase work output, and encourage rhythmic movement (*Nikol et al., 2018*; *Bigliassi et al., in press*; *Karageorghis, Jones & Stuart, 2008*; *Scherer, 2004*; *Terry & Karageorghis, 2011*). The "psychophysical" effects primarily examine the perception of effort which in almost all cases involves the Borgs ratings of perceived exertion scale (RPE) (*Borg, 1982*). These effects are consistent for low and mild intensity activities (*Nikol et al., 2018*; *Terry & Karageorghis, 2011*) and mostly consistent with high intensity activities (*Moss, Enright & Cushman, 2018*; *Haluk, Turchian & Adnan, 2009*).

The effects of music on exercise in the low-to-moderate range of exercise intensities are well established (*Karageorghis & Priest, 2012a*; *Copeland & Franks, 1991*; *Elliott, Carr & Savage, 2004*). Since very high exercise intensities are affected to a high degree by muscle metabolite-induced failures (peripheral fatigue), there is a lesser influence of the central nervous system (central fatigue) compared to lower exercise intensities (*Rejeski, 1985*; *Tenenabum et al., 2004*). However, it has been shown that peripheral fatigue alone is not able to explain the fatigue induced with higher intensity exercise (*Noakes & Gibson, 2004*; *Noakes, 2012*). While fewer studies examined the effect of music on higher intensity exercise, the positive effects seem to persist. For example, listening to music during the Wingate test led to significant improvements in peak power output (PPO) and decreases in fatigue index with the use of music (*Brohmer & Becker, 2006*; *Haluk, Turchian & Adnan, 2009*). This findings are not fully consistent with the perspective of some researchers, suggesting that the "distraction effect" of music is attenuated at higher exercise intensities (>70% maximal oxygen uptake—$\dot{V}O_{2max}$) due to the internal feedback dominating the capacity of the respective afferent nervous system (*Karageorghis et al., 2011*). More importantly, it highlights some gaps in the literature with regard to the so-called intensity limitations of music benefits and the actual mechanisms that result in music ergogenic effects on exercise performance. Further research is still necessary in order to draw decisive conclusions.

Therefore, the primary goal of this study was to examine if listening to high tempo music (130 bpm) while performing high intensity cycling bouts would prolong participants exercise duration and positively effect common physiological measures of fatigue.

## METHODS

### Participants

A convenience sample of 16 healthy and recreationally active individuals (Table 1) volunteered from the university community to participate in a counterbalanced

**Table 1 Subject characteristics.**

|  | Males ($n$ = 8) | Females ($n$ = 8) | All ($n$ = 16) |
|---|---|---|---|
| Age (years) | 30.5 ± 3.7 | 28.3 ± 3.4 | 29.4 ± 3.6 |
| Mass (kg) | 75.2 ± 7.4 | 65.9 ± 4.7 | 70.5 ± 7.6 |
| Height (cm) | 178.3 ± 6.2 | 164.9 ± 5.2 | 171.6 ± 8.8 |
| $\dot{V}O_{2max}$ (L · min$^{-1}$) | 4.1 ± 0.4 | 3 ± 0.3 | 3.51 ± 0.70 |

**Note:**
All data is presented as means and SD.

randomized cross-over design study consisting of a preliminary testing session and two experimental sessions separated by a minimum of 2 days. Recreationally active was defined as an individual who over the last year did not compete in a structured athletic league (i.e., not a varsity athlete) and was active one to three times per week for at least 20 min. All the participants filled out a Physical Activity Readiness Questionnaire from the *Canadian Society for Exercise Physiology (2012)* to determine physical activity level and to screen for a history of cardiovascular, pulmonary, metabolic, and orthopedic conditions. Informed and written consent was sought from study participants. Participants were blind to the hypotheses of the study. The Memorial University of Newfoundland Human Investigations Committee approved the study (IRB approval number: 11.26).

## Experimental design

### Preliminary testing session

Upon the completion of questionnaire filling and anthropometric measurements, participants performed a ramp protocol starting at 50 W at a self-selected cadence (>60 revolutions per minute—RPM) with increment of one W every 3 s to determine $\dot{V}O_{2max}$ and PPO. The test was terminated when the participants reached one of the following criteria: (1) volitional exhaustion, (2) RPE value = 20, or (3) RPM ≤60. The cadence during the incremental test solely assist with monitoring the exercise termination (*Kelly & Basset, 2017*; *Rossiter, Kowalchuk & Whipp, 2006*; *Swart et al., 2009*).

### Experimental sessions

Participants sat on an electronically braked stationary bike with feet secured while wearing an oro-nasal facemask to record cardiorespiratory parameters for the duration of the experiment. The protocol consisted of cycles of 4 min high intensity cycling bouts completed at 80% of PPO (derived from the preliminary testing session) followed by 2 min of active recovery completed at 40% of PPO. This cycle continued until they reached the exercise end-point (more details below). The exercise protocol stems from training practice. Cyclists implement high intensity interval training into the training regimen to improve the maximal aerobic capacity. The 2:1 work/recovery ratio is extensively used by the cycling/running communities and targets the development of endurance characteristics (*Bompa & Haff, 2009*).

Every minute during the 4 min of the high intensity cycling bout, participants were asked to score RPE on the Borg scale, (6–20). After each 4 min high intensity cycling bout,

heart rate (HR) and blood lactate (BL) were recorded while seated on the cycle ergometer. HR was continuously recorded throughout the session using a Polar® HR monitor. Upon exercise termination, four of the five measurements (minus blood pressure) were repeated (see description below). The experimental sessions lasted approximately 2 h each with the ambient conditions being similar across days 24.0 (0.79) °C and 24.0 (0.5) °C, and 1,012.75 (9.7) millibar and 1,015.70 (8.8) millibar for Music and No Music, respectively. Participants were then asked to remain in position and relax for 5 min to ensure proper recording of HR and BL recovery while seated on the cycle ergometer. Blood pressure was measured prior to exercise to ensure no participant was hypertensive. Prior to being released from the experiment, an adequate recovery period (resting HR ≤100 bpm) was given to all participants.

## Instrumentation

### Cardiorespiratory measurements

Oxygen uptake ($\dot{V}O_2$), carbon dioxide output ($\dot{V}CO_2$), breathing frequency and tidal volume were continuously collected with an automated breath-by-breath system (Sensor Medics® version Vmax ST 1.0; Metamax, Yorba Linda, CA, USA) using a nafion filter tube and a turbine flow meter (opto-electric). Respiratory exchange ratio (RER) and minute ventilation ($\dot{V}E$) were calculated as the quotient of $\dot{V}CO_2$ on $\dot{V}O_2$ and as the product of breathing frequency by tidal volume, respectively. HR values were transmitted with a Polar HR monitor (PolarElectro, Kempele, Finland). Prior to testing, gas analyzers and volume were calibrated with medically certified calibration gases (16.0% $O_2$ and 3.98% $CO_2$) and with a three-L calibration syringe. In addition, a propane gas calibration was performed to assess the sensitivity of the oxygen and carbon dioxide analyzers.

### Lactate measurements

All lactate measurements were taken using the Lactate Pro (Arkray KDK, Minneapolis, MN, USA) hand-held portable analyzer. A blood sample of ≥5 μL was taken from the participants fingertip using a spring loaded lancet and then BL values were recorded. The company supplied a check strip to confirm that the analyzer operated correctly, and a calibration strip that provided a non-quantitative indication of analyzer accuracy, which was used at the beginning of each testing session to ensure validity of the measures.

### Cycle ergometer

All exercise protocols were performed on an electronically braked Velotron Dynafit Pro cycle ergometer (RacerMate, Inc., Seattle, WA, USA). Factory calibration of the cycle ergometer was performed using Velotron CS software (RacerMate, Inc., Seattle, WA, USA) and the Accuwatt rundown verification procedure. Individual positional adjustments (saddle and handlebar height) were made before the first exercise test and were replicated for all subsequent exercise tests. Visual feedback of pedalling rate (RPM) was available on the computer monitor to the participants during each exercise session to allow them to monitor intensity and possibly prolong the exercise duration.

*Music*

All participants listened to the exact same playlist of instrumental popular music which was set to a 130 bpm tempo; meaning each song was not originally 130 bpm but was altered to keep a consistent tempo throughout the playlist. We decided on 130 bpm based on previous studies finding that faster tempos lead to superior physical performance (*Waterhouse, Hudson & Edwards, 2010*) and more positive effects compared to slower tempo music conditions (*Edworthy & Waring, 2006*). Music was played through an Ipod® nano using "earbud" type head phones and volume was held constant for each participant at 50% of the maximum volume approximately 65 dB, based on manufacturer specifications of maximum volume being 130 dB.

*Exercise bouts*

Participants were given an explanation of the 6–20 point Borg RPE scale and told that if at any time they wished to stop exercising they could do so. After participants were fitted with an armband that held an Ipod© nano and ear bud type headphones (headphones and armband were worn regardless of condition), the facemask and mesh headpiece were secured next and hooked up to the indirect calorimetric system. Each participant then had a 5 min warm up period where they were instructed to keep a cadence of 60–70 rpm for the duration of the exercise protocol, a parameter displayed on a large computer screen. In both music and no music conditions, the experimental sessions started at 40% of PPO (active recovery), followed by a 4 min high intensity cycling bout at 80% of PPO. The 2:1 ratio was chosen to ensure that participants were able to complete sufficient numbers of exercise bouts and to accumulate enough exercise time for the subsequent analyses. The 2:1 ratio mimics what well-trained cyclists implement in their interval training program (e.g., Fartlek design). Participants were asked to report the RPE score every minute of the 4 min cycling bout. In addition, at the end of the 4 min high intensity cycling bout BL was assessed. Participants repeated the cycle (4 min work load and 2 min active recovery) until reaching one of the following criteria: (1) volitional exhaustion, (2) RPE value = 20, or (3) RPM ≤60. At the precise exercise end-point, time elapsed from the start of the cycling bouts and final RPE values were recorded followed by final BL sample. To avoid confusion, note that for subsequent analyses the exercise duration did not include time spent in active recovery period.

## Data reduction

All data sets were analysed using Sigmaplot (version 10.0; Systat Software Inc., San Jose, CA, USA). First, cardiorespiratory parameters of the incremental test and of the high intensity cycling bouts were smoothed using second-order polynomial function to determine $\dot{V}O_2$, and its corresponding values of $\dot{V}CO_2$, breathing frequency, and tidal volume. Second, HR was time-aligned with the cardiorespiratory parameters and smoothed using the same data reduction technique. Third, RPE scores were interpolated to produce a continuous linear even data point distribution using a two-dimensional interpolation function and were then time-aligned as above-mentioned. Fourth, the high intensity cycling bout epochs were summed to represent time-to-fatigue. Finally, all above

time-aligned parameters were plotted against 25%, 50%, 75%, and 100% of total exercise duration. This strategy was decided upon because the exercise durations of each participant were unknown prior to exercise and expected to differ from one participant and condition to the other. The expression of work intensity in a relative form has definitive merits because it provides the possibility of comparing groups with very different characteristics (*Basset & Boulay, 2000*, *2003*). So, to permit comparison between participants and conditions, we express time in % of total exercise time to reduce inter-subject variability.

## Statistical analysis

All statistical analyses were conducted using Jamovi (version 0.8). Differences between music and no-music conditions in time to task failure (measured in seconds and excluding recovery period) were examined using paired $t$-test. Cardio-respiratory parameters (Bf, $V_T$, $\dot{V}_E$, $\dot{V}O_2$, $\dot{V}CO_2$) and RPE were analyzed with a two-way ANOVA (two conditions × four times (25%, 50%, 75%, and 100% of total exercise duration)). Differences between conditions in blood lactate were measured with a two-way ANOVA (two conditions × three time (pre-test, immediate post-test, 5 min post-test)). HR was analyzed first with a two-way ANOVA (two conditions × four time (25%, 50%, 75%, and 100% of time of failure)). Second, the effect of music upon HR recovery at the same absolute time period for all participants (pre-test and 5 min post-test) was analyzed with paired $t$-tests. Gender was treated as a between-subjects variable in all tests. Differences were considered significant at $p < 0.05$. If significant main effects or interactions were present, a Bonferroni (Dunn) procedure was conducted. Cohen $d$ effect sizes (ESs) were also calculated using the following equation ($d = \frac{\text{Mean differences}}{\text{SD average}}$) in which SD average is $\frac{\sqrt{\text{SD condition 1} + \text{SD condition 2}}}{2}$ to provide qualitative descriptors of standardized effects using these criteria: trivial <0.2, small 0.2–0.5, moderate 0.5–0.8, and large >0.8 (*Cohen, 1988*). Note that since each participant exercised to volitional fatigue, each testing session was a different length of time; therefore, some variables were collapsed over time or reported as a percentage of time-to-fatigue (TTF).

# RESULTS

## Total exercise duration

The average exercise duration in the music condition (10:30 ± 3:38 min:s) was 10.7% longer than the average TTF in the no-music condition (9:33 ± 3:42 min:s) ($p = 0.035$; ES = 0.28). These averages reported times did not include the time spent in active recovery period (Fig. 1A). Therefore on average participants in the music condition completed approximately 2.6 intervals and exercised for 1 min longer while participants in the no-music condition only completed approximately 2.3 intervals.

## Rate of perceived exertion

Small magnitude and non-statistically significant ($p \geq 0.26$, ES = ~0.15) differences were observed between conditions in absolute RPE across the four time points (Fig. 1B).

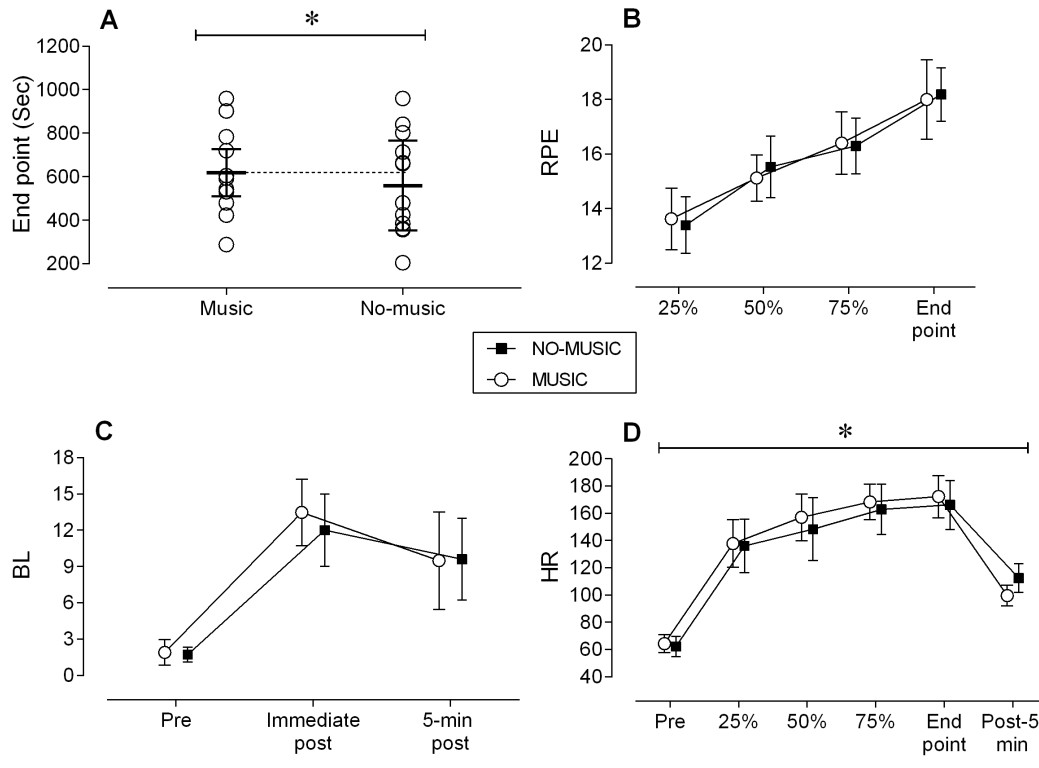

**Figure 1 Performance measures.** All figures include means ± SD. (A) End point (sec). (B) RPE, rate of perceived exertion. (C) BL, blood lactate. (D) HR, Heart rate.

A main effect for time was identified in which RPE increased over the course of the exercise across both conditions ($p < 0.001$).

### Blood lactate

While no statistical interactions or main effect for conditions were observed between conditions at any time point ($p > 0.278$), the average BL levels post-exercise in the music condition at the exercise end point exhibited a moderate magnitude 12.5% higher blood concentration than those at post-exercise in the no-music condition ($13.5 \pm 2.7$ vs. $12.0 \pm 3.0$ mmol $\cdot$ L$^{-1}$; ES = 0.48) (Fig. 1C). These differences likely stem from the longer duration of high intensity activity performed by participants under the music condition.

### Heart rate

First, a small magnitude and statistically insignificant ($p = 0.223$; ES = 0.27) difference was identified between conditions at resting baseline (Music: $66 \pm 6$, No-music: $62 \pm 7$). Second, a main effect for condition was observed in which HR was 4% higher in the music condition across the four time points ($p = 0.043$; ES = 0.25) (Fig. 1D). HR increased in both conditions over time ($p < 0.001$; ES = 1.84). At post 5 min HR was 13% lower in the music condition ($99.6 \pm 7.6$ bpm) than in the no-music condition ($112.6 \pm 10.6$ bpm) ($p < 0.001$; ES = 1.40) (Fig. 1D).

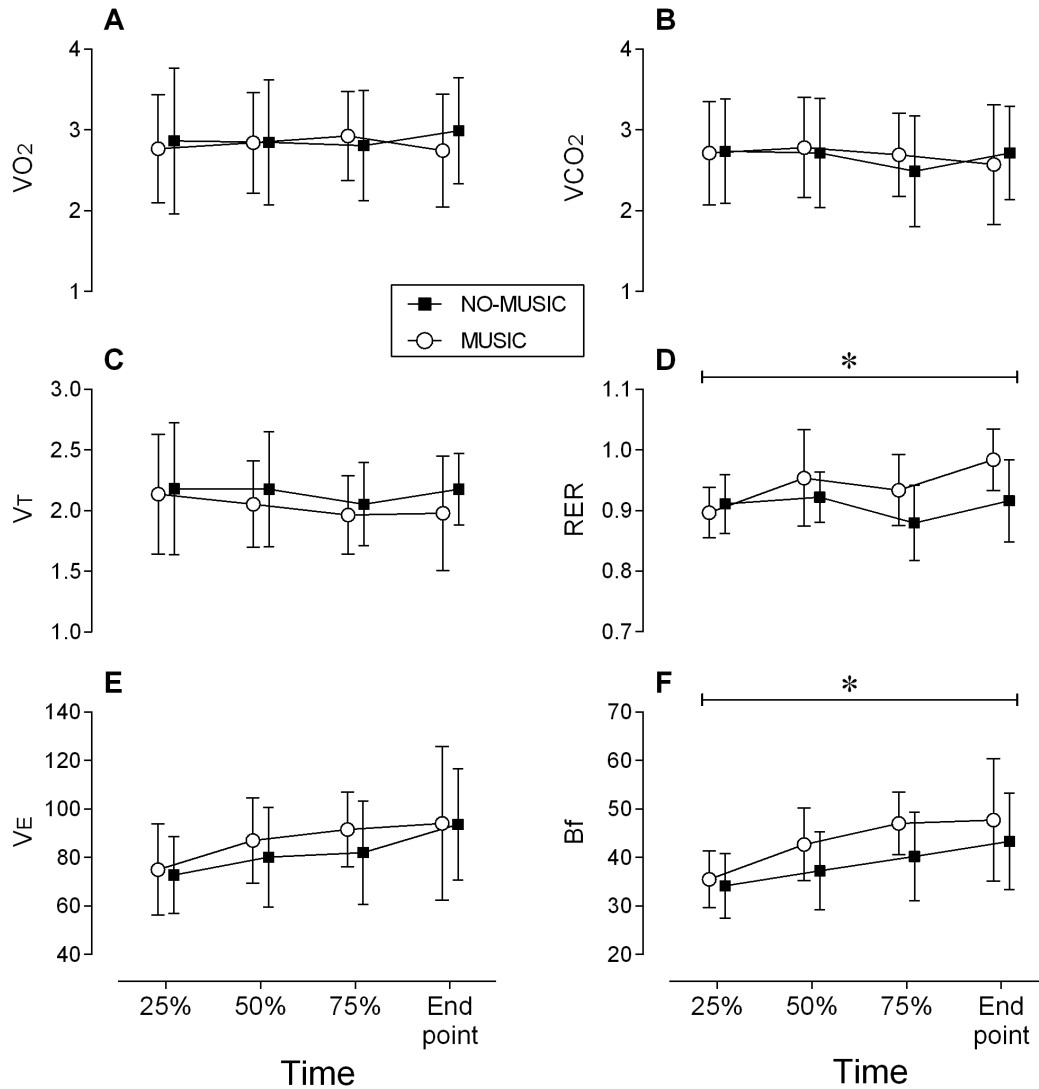

**Figure 2 Metabolic data.** All figures include means + SD. (A) $\dot{V}O_2$, maximal oxygen uptake. (B) $\dot{V}CO_2$, carbon dioxide output. (C) $V_T$, tidal volume. (D) RER, respiratory exchange ratio. (E) $\dot{V}E$, minute ventilation. (F) Bf, breathing frequency.

## $\dot{V}O_2$ and $\dot{V}CO_2$

Small magnitude and statistically insignificant differences were identified in both $\dot{V}O_2$ and $\dot{V}CO_2$ between conditions and across time ($p \geq 0.192$; ES $\leq 0.31$) (Figs. 2A and 2B).

## Ventilation and tidal volume

Small magnitude and statistically insignificant differences were identified in ventilation and tidal volume between conditions ($p \geq 0.012$; ES $\leq 0.27$) (Figs. 2C and 2E). A main effect for time was observed in which ventilation increased from beginning to the end of the cycling bout ($p < 0.001$; ES = 0.91).

### Breathing frequency

A medium magnitude, statistically significant (ES = 0.57; $p$ = 0.006) main effect for condition was observed in which average Bf was 11.6% higher in the music condition (43.2 ± 8.7 breath · min$^{-1}$) compared to the no-music condition (38.7 ± 8.2 breath · min$^{-1}$) (Fig. 2F). A large magnitude and statistically significant main effect ($p$ < 0.001; ES = 1.17) for time was identified in which Bf increased by 23% over time in both conditions from 34.7 ± 6.2 to 45.5 ± 11.5 breath · min$^{-1}$.

### Respiratory exchange ratio

A significant interaction was observed between conditions and time ($p$ = 0.040). However, post hoc testing only revealed statistical differences favoring the music condition at TTF (7%; $p$ = 0.021; ES = 1.1) (Fig. 2D). RER was also 5% greater (non-significant with a large ES magnitude) at 75% of TTF ($p$ = 0.12; ES = 0.85). RER increased over time across both conditions by 5% ($p$ = 0.006; ES = 0.78).

## DISCUSSION

The primary purpose of this study was to determine the effects of listening to high tempo music (130 bpm) on physical performance and acute physiological responses during high intensity interval bouts of cycling. Listening to music led to small increases in total exercise duration, accompanied by an increase in Bf, HR, as well as a steeper HR recovery post-exercise in the high tempo music condition. Yet, RPE scores (Fig. 1B), $\dot{V}O_2$, and $\dot{V}CO_2$ (Figs. 2A and 2B) were similar between conditions although participants exercised longer and recovered faster in high tempo music condition.

A possible explanation for the participants increased exercise duration while experiencing a build-up of peripheral metabolites (increased BL), albeit still reporting comparable RPE scores, can be attributed to a number of psychological influences music had on participants. These include distraction from the sensation of fatigue (*Atkinson, Wilson & Eubank, 2004*; *Bigliassi et al., in press*; *Edworthy & Waring, 2006*), greater arousal and positive affect (*Bigliassi et al., in press*; *Carmichael et al., 2018*; *Yamamoto et al., 2003*; *Schücker et al., 2009*), and better synchronization between music and the motor tasks allowing the activity to be more efficient (*Nikol et al., 2018*; *Rendi, Szabo & Szaba, 2008*; *Waterhouse, Hudson & Edwards, 2010*). Furthermore, models that emphasise the role of the brain in exercise performance and regulation have pointed to the lack of clear peripherally based fatigue reasons to explain why participants reach the point of exercise failure (i.e., inability to continue exercising) since not all muscle fibers are recruited at the point of exhaustion (*Kayser, 2003*; *Noakes, 2000, 2012*). As such, even during high intensity exercise efforts, the brain has a role in regulating effort despite the accumulating metabolites in the muscles (*Billaut et al., 2011*). Henceforth, the CNS must have some control over exercise performance (*Kayser, 2003*; *Noakes, 2000, 2012*).

The increased breathing frequency response of our participants implies that central command cardiovascular drive regulates through medullar integration the dual-talk between the sympathetic and parasympathetic nervous systems even with high intensity exercise. Breathing frequency is controlled via the autonomic nervous system, by the

respiratory centers located in the medulla oblongata in the lower brainstem, functioning largely below the level of consciousness (*Bechbache & Duffin, 1977*; *Williamson, 2010*). Hence, at some point during the cycling intervals, performed with high tempo music, the central neural command seemed to modulate the cardiovascular centers. It is well-known that since the tidal volume plateaus around 60% of $\dot{V}O_{2max}$ any increase in ventilation above this threshold mostly results from breathing frequency (*Folinsbee et al., 1983*; *Jensen, Lyager & Pedersen, 1980*; *Power, Handrigan & Basset, 2012*). However, "it should be clear that no single mechanism can be considered to be the sole mediator of the respiratory response" (*Mateika & Duffin, 1995*). For instance, ventilatory drives due to central command or mechanoreceptor signals, are independent of ventilation, and provide feedforward control of ventilation during constant-load exercise (*Williamson, 2010*). Meanwhile, central command can itself be modified by distractive stimuli (e.g., music; *Boutcher & Trenske, 1990*). Therefore, in this study attention to the external environment might have altered the awareness of physiological sensations bringing about changes in control of ventilation via breathing frequency modulation during high intensity exercise (*Williamson, 2010*). As a result, the participants breathing frequency was higher during high intensity cycling bouts, which could be an indication of the involvement of the cerebral cortex in cardiovascular control mechanisms (*Williamson, 2010*). It should be noted that beside the above-mentioned significant changes in ventilatory response and despite a longer TTF, no other cardiorespiratory parameters were altered by the music condition (Fig. 2). Although the exact mechanism for this physiological response is unclear, these findings do support the idea that further research into the underlying cortical modulation mechanisms involved with the beneficial effects of music on exercise duration are warranted.

Similar to most of the current literature on this topic, HR during exercise was only slightly affected by the music. However, there was an anomaly in the present study, which has to do with post-exercise recuperative effects of music. Participants who completed the cycling intervals with music had a steeper HR recovery. There is very little known about the impact of music on post-exercise performance. While music seems to relieve stress and improve affective states in non-exercise settings (*Särkämö et al., 2008*), participants in this study had the music stimulus removed immediately at the end of high intensity cycling bouts and yet 5 min post-exercise HR were significantly lower compared to no-music condition. It is unclear to us why these large differences occurred. We speculate that music may affect the autonomic nervous system; whereby attention to the external environment seems to reduce the awareness of physiological sensations and negative emotions, which may have triggered a quicker parasympathetic response leading to steeper post-exercise HR recovery. These results further support the theory that the CNS has the ability to control some types of exercise performance and aid in the ability to exercise longer at higher intensities; unfortunately the verification of these theories were not within the scope of this investigation.

Although the research is somewhat conflicting when it comes to measuring the extent to which music can enhance exercise performance at maximal or near-maximal levels, this study demonstrated that listening to high tempo music (via headphones) during high

intensity cycling intervals can lead to greater physiological performance without increasing individual's perceived exertion. The ability to alter perception is consistent with a vast majority of the current research, which is focused on the psychological effects of music on exercise, mood, and emotion and affect. It is the "psychophysical" effects or more prudently the physiological effects that have been noted in this study that should motivate future research endeavours. If music can truly distract or disguise the peripheral signs of fatigue it may increase exercise duration, and perhaps enjoyment and adherence. In view of the results presented here, the central motor drive can be uncoupled from the central cardiovascular command to evoke different circulatory responses (*Williamson, 2010*); indeed, the individuals worked harder while breathing at a higher frequency, and experiencing greater muscle fatigue, all whilst diminishing the feeling of discomfort. The association between central motor drive, central cardiovascular command and perceived exertion was clearly altered by distracting stimuli (high music tempo).

This study suffers from a number of limitations worthy of discussion. First, we altered the tempo of the music to 130 bpm to all songs in the play list. While this allowed for consistency, it may have affected the experience of the listeners. However, instrumental music was used and thus minor changes in tempo would not have had as major an impact as altered music with lyrics. Second, the sample size was based on a convenience sample and not on an a-priori power calculation.

## CONCLUSIONS

The music condition in the present study elicited an increase in exercise duration, breathing frequency, and HR while not influencing RPE and other ventilatory kinetics during exercise. Additionally, a steeper HR recovery post-exercise compared to the control condition was observed. These results support the notion that music can modify the interplay between central motor drive, central cardiovascular command, and perceived exertion. Changes in factors such as breathing frequency suggest that this modification may occur at a subconscious level. Therefore, further studies of the effect of music on exercise performance should focus on the physiological mechanisms responsible for the observed changes, including potentially more complex procedures (i.e., fMRI) as opposed to the psychometric measurements.

### Funding
The authors received no funding for this work.

### Competing Interests
The authors declare that they have no competing interests.

### Author Contributions
- Meaghan E. Maddigan performed the experiments, authored or reviewed drafts of the paper, approved the final draft.
- Kathleen M. Sullivan conceived and designed the experiments, performed the experiments.

- Israel Halperin analyzed the data, prepared figures and/or tables, authored or reviewed drafts of the paper, approved the final draft.
- Fabien A. Basset conceived and designed the experiment, analyzed the data, authored or reviewed drafts of the paper, approved the final draft.
- David G. Behm conceived and designed the experiments, analyzed the data, contributed reagents/materials/analysis tools, authored or reviewed drafts of the paper, approved the final draft.

## Human Ethics

The following information was supplied relating to ethical approvals (i.e., approving body and any reference numbers):

The Memorial University of Newfoundland Human Investigations Committee approved the study (IRB approval number: 11.26).

## Data Availability

The raw data is provided in the Supplemental Files.

## Supplemental Information

Supplemental information for this article can be found online at http://dx.doi.org/10.7717/peerj.6164#supplemental-information.

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
