# Peer review of "High tempo music prolongs high intensity exercise"

_PeerJ, doi:10.7717/peerj.6164_

## Round 0.1 · original submission · Major Revisions

I would like to invite the authors to revise their manuscript according to the reviewers' comments.

·

Basic reporting

no comment

Experimental design

no comment

Validity of the findings

no comment

Additional comments

Lines 108-109: Please provide references for the termination of the protocol.
Line 187-188: You mentioned gender as a between-subject factor but the potential effects of baseline VO2 consumption was not taken into account. It might be possible that longer exercise duration was because the subjects were more prepared.
Line 145: You mentioned that you altered the tempo to be consistent throughout the playlist. While it might be justifiable to standardize your protocol, it could have affected the experience of listening to music (specially when it includes lyrics). Please explain this further as a limitation.

- Please discuss the clinical significance of measuring ventilation, tidal volume, and RER in your study.
- Please mention whether the number of participants was based on a pilot study (power analysis) and if not, mention it as a limitation and provide suggestions for future studies.

Reviewer 2 ·

Basic reporting

Language is scientific but authors are advised to use specific terms as mentioned in comments. Sufficient Literature references and field background/context provided by authors.
Corrections advised in Fig 1 A as it is incomplete.
Otherwise complete.

Experimental design

What make you decide music tempo 120 bpm was high for subjects ? Have you followed any classification? Kindly cite reference.
Please provide basic anthropometric characteristics of the study subjects if noted any.

Line 45- use more specific term for "as an external distracting stimulus during exercise" as Dissociation.

Line 55-56: The third explanation postulates that during continual submaximal activity, anindividual is predisposed to respond to rhythmical elements (music being one of many rhythmical patterns); cite reference.

Line 60-65: The available evidence on this topic is congruent and demonstrates that music can and does have a consistent and measurable effect on attention, the ability to trigger a range of emotions, affect mood, increase work output, and encourage rhythmic movement (Atkinson et al. 2004; Karageorghis 2008; Scherer 2004; Terry and Karageorghis 2011; Yamashita et al. 2006). The ‘psychophysical’ effects primarily examine the perception of effort (Pandolf 1978), which in almost all cases involves the Borg’s Ratings of Perceived Exertion scale (RPE). Emphasis effect of music as mentioned above—does it correlates with high intensity exercise as well or is just for mild / moderate intensity exercise only.

Methodolgy:
Line92 Kindly state what do you mean by recreationally active individual ? State whether study subjects are trained / untrained/ ameteurs. Kindly give how much amount of physical activity per day/per week/per month and how long were engaged before undergoing testing.

Line 97 – Any reference given in observation section regarding Physical Activity Readiness Questionnaire from the Canadian Society for Exercise Physiology to determine physical activity level and to screen for a history of cardiovascular, pulmonary, metabolic and orthopedic conditions.

Line 100 – informed and written consent was sought from study subjects.
Methodology section haven’t elaborated the method for estimation of VO2max .
Similarly recording of basic parameters like HR, BP at resting / baseline level were recorded in which position of body whether supine or sitting position.

LINE 108:
The test was terminated when the participants reached one of the following criteria: 1) volitional exhaustion, 2) RPE value ≥ 19, or 3) RPM ≤ 60. What was the maximumcadence value reached during testing session in both males and female?
Reason for having pretesting session for the study subjects. Kindly justify.

LINE 116 :
Upon exercise termination, four of the five pre-test measurements (minus blood pressure) were repeated (See description below). Why BP wasn’t taken into account post exercise.Do all subject undergo test session otn the very same day? How much was the study duration?What were the ambient conditions during high intensity exercise testing?

LINE 143:
Please elaborate –Purpose of visual feedback of pedalling rate (RPM)

LINE 144:
Does playlist implies same song been played for all participants or different song from the playlist ?

LINE 154 :
During warm up parameters were noted or not?

LINE 158-159:
Were subject informed and explained about reporting of RPE score before initiating testing?

LINE 161:
Kindly explain “Active recovery” Why 2 miunutes time duration wass expected for active recovery with high intensity workout? Explain.
Exercise testing under both that is test and control condition for the same subjects were done on the same day or different days ? If done on same day how much interval was there in between two sessions.

LINE 211 :
What may be possible explaination for raised BL levelsin music condition than without music condition? Baseline/resting HR, BL levels weren’t compared with post exercise levels.

OBSERVATION:

Changes in BP levels wasn’t noted- kindly explain.
LINE 252: Does CNS regulation impacts exercise performance? What could be putative factor which increase TTF in high intensity exercise setup?

FIG 1 A No music condition in TTF graph isn’t represented correctly.

It would have been better if anthropometric data was also if represented in graphical form .

Validity of the findings

No comment

Additional comments

Some corrections suggested with relevant explanation as asked in experimental designed from the authors .

Annotated reviews are not available for download in order to protect the identity of reviewers who chose to remain anonymous.

·

Basic reporting

Writing quality is mostly very good.
Literature references generally seem very appropriate, though perhaps more recent references can be added if available.
Structure of the paper seems appropriate.

Experimental design

Design is good.
I am unsure about the way in which percentage of time to completion was handled (see comments below).

Validity of the findings

Findings seem valid.

Additional comments

Comments

I very much like the idea behind the study and feel as though the content is timely and appropriate for the journal.

The manuscript is written well and is easy to follow.

The authors are to be commended for writing quality.

Abstract

Some mention of the sample seems appropriate.

Lines 34-36: Perhaps note the point in time when these differences were observed. Likely at or near the end of exercise but this should be stated more explicitly.

Line 38: Not stated is whether music continued during the recovery. This is important.

Introduction

Line 52: Using the word ‘somehow’ seems to beg for an explanation. Provide the explanation perhaps or edit the wording.

Line 65: A citation for the Borg scale should be provided.

Line 66: It seems odd to refer to the effect of music as a trend in the literature. Consider editing to create greater precision.

Line 76: The studies noted as more recent are not in fact more recent than the manuscript suggests. In the next statement you seem to suggest that more recent studies (06 and 09) now contradict conclusions made in the 2011 paper. My request here is that you more carefully consider the timing of publications and related statements so as to most accurately reflect the literature. And the following statement related to the 2011 publication suggests

Method

Line 106: The specifics of the cycle ergometer are described later in the manuscript but the mention here of self-selected cadence immediately left me wondering whether the bike was braked electronically or mechanically. If an adjustment here can be reasonably made please consider it.

Line 107: The language of this statement seems to suggest that the workload changes of the test are somehow specifically linked to VO2max determination. I of course know what you mean but more exacting language may be beneficial.

Line 109: It seems odd that the test would perhaps automatically terminate at an RPE of 19 when an RPE of 20 may have been achievable with additional time on the test. Also, it seems appropriate to provide justification and perhaps a citation for the stated termination criteria.

Line 110: The description of the experimental sessions leaves me somewhat unclear. The abstract notes the trials are ‘time to task failure’ but the methods describe a 4-minute test. This seems contradictory, or at least requires greater clarification.

General Note: The many uses of ‘4-min’ and ‘5-min’ seem inconsistent with traditional grammar/writing rules.

Line 126: The language ‘quotient of VCO2 on VO2’ seems awkward.

Line 129: Check spacing around gas percentages.

Line 138: The braking mechanism for the ergometer should be specified.

Line 141: Perhaps the procedure should be cited.

Line 160: Some justification of the work and recovery intensities and durations should be provided.

Line 161: The cycling protocol is described more clearly here than before, but edits to prior mentions of the protocol remain appropriate.

Line 161: Designation of the trial as being linked to ‘failure’ seems inappropriate if in fact the trial can be terminated by RPE values only, especially if a value of 19 out of 20 would be a reason to terminate the test. This seems like an important matter. I am curious how many trials ended because of RPE. If relatively few then perhaps those trials could be deleted from analyses.

Line 162: If I am understanding the design properly a participant who stops perhaps 15 seconds before the conclusion of one of the 4-min intense segments in comparison to a participant who makes it through those final 15 seconds and the recovery that follows and likely the first 60 plus seconds of the subsequent 4-minute interval would have dramatically different ‘time to failure’ but not necessarily very different capacities. Really, their capacities would likely be far more similar than the time difference would indicate. Some justification for the 4-min hard and 2-min easy approach to the protocol seems useful.

Lines 182 & 185: It is not clear to me how the 4 data collection points for the time variable would necessarily equate to specific percentages of completion because time to failure would not be known in advance of starting the trial.

Note: A justification of sample size should be provided.

Results

Line 199: This disclaimer should maybe be provided earlier in the manuscript because I was not thinking this was the case when considering the time figures in the previous sentence.

Line 205: Again, how would the time points be known in advance? That is, the 75% time mark would be different for a trial that lasted 8 mins in comparison to a trial that lasted 11 mins.

Line 215: The p-value and ES value seem rather incongruent. Some of the difference is likely linked to the small sample size limiting the likelihood of statistical significance but a .5 ES would seem likely to produce a more favorable p-value. Please consider a careful inspection of the data to seek a better understanding of why this may have occurred.

Line 217: Reporting actual HR values more in this section would be helpful.

Line 221: The reported HR difference in recovery are really large and almost unexplainable. Please consider working towards developing a defensible position on this difference.

General Note: I don’t think the literature review and introduction of the research idea provide a compelling case for why the Ve, TV, and Bf are included in the analyses. Reporting Ve seems more reasonable and appropriate as a marker of overall ventilator work but inclusion of TV and Bf in particular seem unnecessary without greater context and justification.

Discussion

Line 253: The lack of a significant difference in RPE is perhaps expected because music likely created a distraction that impacted ratings in a predictable manner.

Line 268: I am not an expert on cardiovascular control but it seems that the manuscript is suggesting that TV and Bf are under separate control mechanisms. If this is the case it seems more coverage of this topic is warranted in the manuscript than is currently provided, both to address these findings and more generally to help the reader better understand the context of the research design.

Line 277: The post-exercise HR values are very surprising, especially as it relates to the magnitude of the difference and because the music was removed immediately upon completion of exercise. My lack of expertise on neurophysiology of the heart limits my ability to properly evaluate this particular aspect of the design and related findings but I am drawn to the possibility that this finding should be explored a bit more.

Line 292: The word ‘effort’ may not be appropriate given that RPE is a dependent variable within the study. Maybe ‘physiological performance’ would be better.

Conclusions

Line 306: Perhaps also mention other dependent variables that did not differ, not just RPE.

References

Seem good and appropriate.

Figures

Figure 1: Check spelling of ‘measures’ at the top of the figure.

Figure 1: The notation of variable should include the ‘minus’ of the SD not just the plus.

---

## Round 0.2 · Minor Revisions

I would like to thank the authors for providing the detailed responses to the reviewers and adequately address the points indicated during the peer review process. Please note there still are some minor comments to be addressed. Therefore, I invite the authors to proceed with the edits before having the final recommendation.

·

Basic reporting

no comment.

Experimental design

This is not an original study.

Validity of the findings

no comment.

Additional comments

Thank you for your revision.
I am still not convinced about why you compared ventilation, tidal volume, and RER, and not other respiratory parameters. Please try to justify them in your discussion.

Reviewer 2 ·

Basic reporting

All reference as suggested earlier are cited.
Explanations asked were submitted and found to be rational.
Earlier comment regarding Fig 1 A withdrawn. Fig I1 A is appropriate.

Experimental design

Yes the article is within scope of the journal.
Research questions were defined properly.
Methods described are appropriate.

Validity of the findings

No comments

Additional comments

Research article can be accepted as the corrections and changes have been incorporated by the author with explainations.

·

Basic reporting

Appropriate.

Experimental design

Reasonable.

Validity of the findings

Good.

Additional comments

The authors are commended for their excellent work in attending to the many comments provided by the various reviewers. Any remaining concerns on my part are not particularly substantive.

---

## Round 0.3 · accepted · Accept

We appreciate the efforts made to improve the paper during the peer review.

#